# The Complete Chloroplast Genomes of Three *Manglietia* Species and Phylogenetic Insight into the Genus *Manglietia* Blume

**DOI:** 10.3390/cimb47090737

**Published:** 2025-09-10

**Authors:** Yuan Luo, Wei Luo, Tongxing Zhao, Jing Yang, Lang Yuan, Pinzheng Zhang, Zixin Gong, Haizhu Li, Yongkang Sima, Tao Xu

**Affiliations:** 1School of Ecology and Environmental Science, Yunnan University, Kunming 650504, China; ly15699002997@163.com (Y.L.); lw754087@163.com (W.L.); 19533666030@163.com (T.Z.); yangjing202306@126.com (J.Y.); yl18788488061@126.com (L.Y.); 18630955550@163.com (P.Z.); gong54257710@163.com (Z.G.); lhz6573679@163.com (H.L.); 2Kunming Arboretum, Yunnan Academy of Forestry & Grassland Science, Kunming 650201, China; simayk@163.com

**Keywords:** *Manglietia*, chloroplast genomes, interspecific relationship, phylogeny

## Abstract

*The genus Manglietia Blume* is an important group of Magnoliaceae that has high economic and ornamental value. Owing to the small morphological differences among most *Manglietia* species and the limited sample sizes in previous molecular-level studies, its infrageneric classification remains unclear, and interspecific relationships for some species are still contentious. Clarifying the phylogenetic relationships within the genus *Manglietia* is crucial for species classification, genetic diversity assessment, and evolutionary developmental studies. This study sequenced, assembled, and annotated the chloroplast (cp) genomes of *Manglietia guangnanica*, *Manglietia hookeri*, and *Manglietia longirostrata*. The results indicated that these cp genomes are canonical quadripartite structures with total lengths of 160,067 bp, 160,067 bp, and 160,076 bp, respectively. All three cp genomes were annotated with 133 genes, comprising 88 protein-coding genes, 37 tRNAs, and 8 rRNAs. A total of 31, 30, and 30 dispersed repeats and 53, 53, and 56 SSRs were detected, respectively. ENC plot, neutrality plot, and PR2 plot analyses indicated that codon usage bias was influenced primarily by natural selection. Nucleotide diversity analysis revealed 8 highly variable regions in the cp genomes, among which *petA*-*psbJ*, *rpl32*-*trnL*, and *ccsA*-*ndhD* are recommended as candidate molecular markers for *Manglietia* species. Phylogenetic analysis revealed four highly supported clades: Clade I (18 species), Clade II (*M. decidua* only), Clade III (9 species), and Clade IV (*M. caveana* only). Among these clades, Clade IV is a newly discovered monotypic clade in this study, which differs from the results of all previous studies. Further investigations of Clades I and III, which include multiple *Manglietia* species, revealed that the presence or absence of hairs on Twigs, Stipules, and the abaxial surface of the leaf are important morphological characteristics for distinguishing species between these two clades. Furthermore, the results revealed that *M. guangnanica* and *M. calcarea* are two distinct species, and the treatment of *M. longirostrata* as a variety of *M. hookeri* was not supported by our study. This study enriches the cp genome data of *Manglietia*, provides new insights into infrageneric classification, and lays a foundation for further phylogenetic and evolutionary studies of *Manglietia.*

## 1. Introduction

Chloroplasts are important organelles in plant cells; they possess a complete genome and are capable of independent transcription and translation. Compared with the nuclear genome, the plant chloroplast (cp) genome is smaller, structurally more stable, presents moderate mutation rates, and is predominantly uniparentally inherited in angiosperms [1]. These characteristics make it more advantageous for species identification and the study of phylogenetic evolution [2]. In the past, researchers often used single genes or segments composed of multiple genes to study phylogenetic relationships. For example, *matK* and *ndhF* in the cp genome have been demonstrated to be important molecular markers for exploring phylogenetic relationships within Magnoliaceae [3]. However, limited segments often carry only partial genetic evolutionary information, thus failing to fully reveal phylogenetic relationships among groups. In recent years, the rapid development of biotechnology has led to a reduction in the cost of high-throughput sequencing, prompting researchers to increasingly favor the use of cp genomes, which contain complete genetic information, as molecular evidence for phylogenetic inference [4]. Wang et al. conducted a phylogenetic analysis using cp genome sequences from 86 Magnoliaceae species and proposed a classification system for the Magnoliaceae comprising 2 subfamilies, 2 genera, and 15 sections [5]. Through the investigation of complete cp genomes from three *Houpoea* species, Xu et al. clarified the infrageneric relationships within the genus [2]. Wu et al. reported on the cp genomes of eight *Lirianthe* species and, combined with morphological analysis, clarified the subgeneric classification relationships within the genus *Lirianthe* Spach, further resolving the issue of its classification within the Magnoliaceae [6]. Numerous studies utilizing cp genomes as molecular evidence to address phylogenetic relationships among different Magnoliaceae species and intrafamily phylogenetic relationships are being reported.

The genus *Manglietia* Blume was first established by Blume [7]. This genus comprises approximately 40 species of arboreal trees worldwide, which are primarily distributed in tropical and subtropical regions of Asia, with the highest diversity found in subtropical regions [8]. China has approximately 27 species, which are primarily distributed in the southwest and south regions [9]. Most species in this genus have straight and tall trunks, esthetically pleasing wood grains, dense foliage, and large, fragrant flowers. They are not only excellent timber sources for construction and furniture but also important ornamental trees in landscaping. Furthermore, parts of some species (such as *M. fordiana* Oliv.), including their fruits and bark, can be used medicinally to treat constipation and dry coughs, indicating significant economic and ornamental value [10]. The first classification of *Manglietia* was proposed by Tiep [11]. On the basis of the ratio of style length to carpel length, he divided the genus into two sections: Sect. *Manglietia* (ratio > 1/2) and Sect. *Oliveria* N.V. Tiep. (ratio < 1/2). In 1995, Zheng classified *M. decidua* Q.Y. Zheng into *Manglietia* on the basis of morphological characteristics [12]. He further divided *Manglietia* into Sect. *Manglietia* (evergreen habit) and Sect. *Decidua* Q. Y. Zheng (deciduous habit) on the basis of growth habit, with Sect. *Decidua* Q. Y. Zheng containing only *M. decidua*. The characteristic of the style being no longer than half the length of the carpel became the diagnostic feature for *M.fordiana*, and the systematic position of *M. decidua* within *Manglietia* was acknowledged. However, the classification systems proposed by both researchers were not adopted by researchers. Xia subsequently divided the 27 species of *Manglietia* into two subgenera, subgen. *Manglietia* and Subgen. *Sinomanglietia* (Z. X. Yu) N.H. Xia, which is based on the dehiscence mode of mature carpels and the torus. He further subdivided Subgen. *Manglietia* can be divided into two sections, Sect. *Manglietia* and Sect. *Coniferae* N. H. Xia, on the basis of pedicel length [13]. However, on the basis of long-term observations of living plants, Sima argued that pedicel length can vary considerably even within the same species. Classification on the basis of this characteristic might separate different populations or individuals of the same species into different groups [9]. Therefore, they divided the 27 species of *Manglietia* into two sections, Sect. *Manglietia* and Sect. *Deciduae* Q. Y. Zheng “*Decidua*” (containing only *M. decididua*), on the basis of characteristics such as the dehiscence mode and fusion state of mature carpels and the type of indumentum (especially trichome stalks) on the abaxial surface of the leaf. Sect. *Manglietia* is further divided into two subsections: Subsect. *Metamanglietia* Sima & S. G. Lu, and Subsect. *Manglietia*. In 2024, Li et al. divided 19 species of *Manglietia* into three clades on the basis of a phylogenetic study of 77 protein-coding genes common to Magnoliaceae cp genomes [14]. In their study, *M. decidua*, the only deciduous species in *Manglietia*, was placed on a separate branch, which is consistent with the morphological classification results of Zheng and Sima. However, the phylogenetic relationships of *Manglietia* species and the genus-level grouping system proposed still differ from those of previous studies. Although some research on *Manglietia* has been conducted, due to the limited number of collected samples, the genus-level grouping of *Manglietia* remains unclear, and there are still discrepancies in interspecific relationships. Therefore, a more comprehensive study of the infrageneric phylogenetic relationships within *Manglietia* is still needed.

Since most species of *Manglietia* share similar morphological characteristics, combining morphological and molecular data to establish a reasonable interspecific relationship is necessary. However, many recently described new species often lack molecular-level studies, which contributes to unresolved questions regarding the relationships among certain species in the genus *Manglietia*. *M. longirostrata* (D.X. Li & R.Z. Zhou ex X.M. Hu, Q.W. Zeng & L. Fu) Sima and *M. guangnanica* D. X. Li & R. Z. Zhou are both recently described species within the genus *Manglietia* and are excellent ornamental plants. However, owing to human-induced environmental changes and deforestation, their wild populations are extremely rare. *M. longirostrata* is distributed only in Malipo County, Yunnan Province, China. At the time of its discovery, only three individuals existed, with seedlings being scarce. In 2012, Hu et al. observed the sample and concluded that it was morphologically similar to *M. hookeri* Cubitt & W. W. Smith and published it as a variety of *M.hookeri* (*M.hookeri* var. *longirostrata* D.X. Li & R.Z. Zhou ex X.M. Hu, Q.W. Zeng & L. Fu) [15]. Later, after extended observations, Sima et al. argued that *M. longirostrata* differs significantly in morphology from *M. hookeri* and other species within *Manglietia* and should be recognized as a distinct species [16]. Consequently, the systematic position of *M. longirostrata* within *Manglietia* has been debated: whether classifying it as a variety of *M.hookeri* or as an independent species is more reasonable remains unresolved and requires further study. *M. guangnanica* is found only in Guangnan County, Yunnan Province, and the southwestern region of Guizhou Province, China [17]. Hu et al. described it as a distinct species under *Manglietia* on the basis of morphological observations and noted that *M. guangnanica* shares morphological similarities with *M. pachyphylla* Huang T. Chang, *M. crassipes* Y. W. Law, and *M.fordiana* Oliver, suggesting a close phylogenetic relationship [18]. Subsequently, during further specimen consultation for this study, databases such as GBIF Backbone Taxonomy (https://doi.org/10.15468/39omei Accessed via https://www.gbif.org/species/11026937 accessed on 20 July 2025), POWO (https://powo.science.kew.org/ accessed on 20 July 2025), and WCVP [19] all treat it as a synonym of the *M. calcarea* X. H. Song. Furthermore, when submitting the cp genome information of *M. guangnanica* to the NCBI database, they named it *M. fordiana* var. *calcarea* (X. H. Song) B. L. Chen & Nooteboom. Owing to its recent description, studies on *M. guangnanica* have thus far covered only morphology, palynology, and floristics and lack phylogenetic investigations. While the placement of both *M. guangnanica* and *M. longirostrata* within the genus *Manglietia* is unquestionably accepted, their infrageneric systematic positions and interspecific relationships remain contentious. To further clarify the phylogenetic relationships within *Manglietia*, molecular-level studies on *M. guangnanica* and *M. longirostrata* are of paramount importance.

This study employed high-throughput sequencing technology to sequence *Manglietia guangnanica*, *Manglietia longirostrata*, and *Manglietia hookeri*. Through assembly and annotation, we obtained the complete cp genome sequences for these three *Manglietia* species. The primary objectives are as follows: (1) to present the complete cp genome sequences and characterize their features for these three *Manglietia* species; (2) to clarify the interspecific relationships between *M. longirostrata* and *M. hookeri*, as well as between *M. guangnanica* and *M. calcarea*; and (3) to investigate the infrageneric phylogenetic relationships within *Manglietia* on the basis of a larger number of species, with the aim of establishing a more reasonable and natural infrageneric classification system for the genus.

## 2. Materials and Methods

### 2.1. Sample Collection, DNA Extraction, and Sequencing

For this study, *M. guangnanica*, *M. longirostrata*, and *M. hookeri* plant material was collected from the Yunnan Academy of Forestry and Grassland Science (Kunming, China), with voucher specimen numbers M00100683, M00100187, and M00100177, respectively. The species was determined by Sima Yongkang. Fresh, young leaves were collected, immediately placed in collection bags upon harvest, and stored at −20 °C for preservation. Total genomic DNA was extracted from fresh leaf tissue via the Plant Genomic DNA Rapid Extraction Kit (Sangon Biotech, Shanghai, China) following the manufacturer’s protocol. Agarose gel electrophoresis was used to assess DNA integrity (concentration: 1%, voltage: 200 V, time: 30 min), and Qubit (Thermo Fisher Scientific, Waltham, MA, USA) was used to quantify DNA sample concentration. Qualified DNA was used for PE library (150 bp) construction and sequenced on the HiSeq platform (Illumina, San Diego, CA, USA). At least 5 Gb of clean data were generated for each species.

### 2.2. Chloroplast Genome Assembly and Annotation

GetOrganelle v1.7.7.0 [20] was used to filter paired-end reads from the clean data and assemble them into contigs, with parameters -k 21, 77, 127. The assembled cp genomes were manually inspected in Bandage and subsequently edited in Geneious v2024.0 [21] to obtain complete plastid genomes for each sample. Using *M. longirostrata* (MT584886) as the reference sequence, annotation was performed with the Plastid Genome Annotator (PGA) v2.0 [22], followed by manual verification and refinement in Geneious v2024.0 as needed. The cp genome information of the three species was subsequently submitted to NCBI, with accession numbers as follows: *M. guangnanica* (PQ246290), *M. hookeri* (PV239602), and *M. longirostrata* (PV239601). This yielded three high-quality chloroplast genomes. The structural features of the cp genomes of all three species were then visualized via OGDRAW v1.3.1 [23].

### 2.3. Repeat Sequence Analysis

The online tool REPuter (https://bibiserv.cebitec.uni-bielefeld.de/reputer/ accessed on 1 July 2025) [24] was used to detect dispersed repetitive sequences, including forward repetitive sequences, reverse repetitive sequences, palindromic repetitive sequences, and complementary repetitive sequences. The parameter settings were as follows: the minimum repeat length was restricted to 30 bp, the sequence identity was ≥90%, and the Hamming distance was set to 3. Tandem repeats were identified via Tandem Repeats Finder (TRF) [25]. The MISA v2.1 [26] tool can be used to identify simple sequence repeats (SSRs), with parameters set to 10, 5, 4, 3, 3, and 3.

### 2.4. Analysis of Codon Usageis

Protein-coding sequences (CDSs) were extracted from the three *Manglietia* cp genomes via PhyloSuite v1.2.3. [27] CDSs were filtered according to the following criteria: length ≥ 300 bp, presence of a canonical start codon (ATG), and absence of internal stop codons. CodonW v1.4.2 (https://sourceforge.net/projects/codonw/ accessed on 1 July 2025) was subsequently used to calculate the relative synonymous codon usage (RSCU) and effective number of codons (ENC) values of protein-coding sequences. The GC content at the first (GC1), second (GC2), and third (GC3) codon positions was computed via EMBOSS v6.6.0 [28]. Subsequently, ENC and GC3 were used for ENC-polt analysis, GC12 and GC3 were used for neutrality plot analysis, and the percentage of each base at position 3 of the codon was used for PR2 plot analyses.

### 2.5. Genomic Comparative Analysis

To elucidate interspecific relationships and further investigate divergence among the *Manglietia* cp genomes, we conducted comparative analyses of three newly sequenced *Manglietia* species with three additional species whose cp genomes. First, CPJSdraw v1.0 [29] was employed to visualize expansions and contractions at inverted repeat (IR) boundaries. MAUVE v2.4.0 [30] was subsequently used to analyze collinear relationships across the six *Manglietia* cp genomes. Finally, whole-genome alignments were performed via mVISTA [31] in Shuffle-LAGAN mode, with *M. guangnanica* serving as the reference sequence.

### 2.6. Ka/Ks Ratio Analysis

To compare evolutionary divergence in coding regions across the six *Manglietia* cp genomes, selective pressure analysis was performed. Protein-coding genes were extracted from the six cp genomes via PhyloSuite v1.2.3. To enhance reliability, the extracted genes were filtered according to the aforementioned criteria (length ≥ 300 bp, canonical ATG start codon, no internal stop codons). The nonsynonymous substitution rate (Ka), synonymous substitution rate (Ks), and Ka/Ks ratios were then calculated for protein-coding genes via KaKs_calculator v2.0, employing the MLWL model [32].

### 2.7. Nucleotide Diversity Analysis

To identify hypervariable regions in the cp genomes of *Manglietia* species, we acquired 31 cp genomes comprising three newly sequenced genomes in this study and 28 publicly available genomes downloaded from the NCBI database. The sequences were aligned via MAFFT v7.490 [33], followed by nucleotide diversity calculation via DnaSP v6 [34] by sliding window analysis (window size = 600 bp, step size = 200 bp).

### 2.8. Phylogenetic Trees Construction and Analysis

To resolve the infrageneric phylogenetic relationships within *Manglietia*, the cp genome sequences of two *Houpoea* species were downloaded from the NCBI database as the outgroup. The accession numbers for all included species are listed in Appendix A. Phylogenetic trees were reconstructed via both maximum likelihood (ML) and Bayesian approaches using protein-coding sequences (CDSs) shared across 33 cp genomes. The GenBank annotation files of the 33 cp genomes were visualized via Geneious v2024.0 software and manually checked and adjusted to increase the reliability of the phylogenetic analysis results. PhyloSuite v1.2.3 was used to extract 85 protein-coding sequences shared by the 33 cp genomes. These sequences were aligned with MAFFT v7.490, followed by trimming of low-quality alignment regions via trimAl v1.4.1 [35]. Finally, PhyloSuite v1.2.3 was used to concatenate the 85 protein-coding genes to construct the phylogenetic tree. ModelFinder was used to determine the optimal model for constructing the phylogenetic tree [36]. The ML tree was constructed with RAxML 8.2.11 [37] under the HKY85+GAMMA model with 1000 bootstrap replicates. The Bayesian tree was generated via MrBayes 3.2.6 [38] under the GTR+GAMMA model, with Markov chain Monte Carlo (MCMC) parameters set to 1,000,000 generations and 4 chains and a burn-in fraction of 0.25. Final tree visualization was conducted in iTOL v5 (https://itol.embl.de/ accessed on 25 July 2025).

## 3. Results

### 3.1. Genome Size and Basic Structural Features

The cp genomes of *M. guangnanica*, *M. longirostrata*, and *M. hookeri* are 160,067 bp, 160,076 bp, and 160,067 bp in length, respectively. All three exhibit a typical quadripartite structure, consisting of two single-copy regions (LSC and SSC) and a pair of inverted repeats (IRa and IRb) (Figure 1). For the three *Manglietia* species, the lengths of the LSC regions are 88,107 bp, 88,095 bp, and 88,125 bp; the SSC regions are 18,807 bp, 18,837 bp, and 18,776 bp; and the IR regions are 26,576 bp, 26,572 bp, and 26,583 bp, respectively. The total GC content was identical for all three samples, at 39.3%. Furthermore, the arrangement and order of genes are identical across the three *Manglietia* cp genomes. A total of 131 genes were annotated in each genome, comprising 86 protein-coding genes, 37 tRNA genes, and 8 rRNA genes (Table 1). In the cp genome, 19 genes, including *rpl2*, *rps7*, and *rrn4.5*, have two copies; 14 genes, including *rps16*, *rpoC1*, and *petB*, contain one intron; and the *rps12*, *clpP*, and *ycf3* genes contain two introns (Table 2).

### 3.2. Analysis of Repeats and SSRs

The repeat sequences were categorized into four types: forward repeats (F), reverse repeats (R), complementary repeats (C), and palindromic repeats (P). This study analyzed long repeat sequences in the complete cp genomes of three *Manglietia* species. The results indicate that *M. guangnanica*, *M. longirostrata*, and *M. hookeri* contain 31, 30, and 30 dispersed repeats, respectively. Among these species, only forward and palindromic repeats were detected, with no reverse or complementary repeats identified. There were slightly more palindromic repeats than forward repeats in *M. guangnanica* and *M. hookeri*, whereas forward repeats were slightly more common in *M. longirostrata*. All long repeats in the three *Manglietia* species exceeded 30 bp in length, with the majority concentrated within the 30–44 bp range. Repeats in the 60–74 bp range were absent (Figure 2).

Using MISA, we detected 53, 53, and 56 simple sequence repeats (SSRs) in the cp genomes of *M. guangnanica*, *M. hookeri*, and *M. longirostrata*, respectively. The SSR types identified in all three species included mono-, di-, tri-, tetra-, hexa-, and compound nucleotide repeats. No pentanucleotide repeats were observed, with mononucleotide repeats being the predominant type (Figure 3C). The di- and trinucleotide repeats were exclusively distributed in the LSC region, whereas the hexanucleotide repeats occurred only in the IR regions. Additionally, A and T bases not only dominated mononucleotide repeats but also presented high abundances of di-, tri-, tetra-, hexa-, and compound nucleotide repeats (Figure 3A). Among the cp genomes of the three *Manglietia species*, SSRs were located predominantly in the LSC region (39, 39, 43) and least frequently in the IR region (4, 4, 4) (Figure 3D). As shown in Figure 3B, most SSRs were located in intergenic regions (IGRs) (38, 38, 39). Compared with those in *M. guangnanica* and *M. hookeri*, the SSRs in the *M. longirostrata* cp genome presented certain differences in quantity, type, and distribution patterns.

### 3.3. Codon Usage Bias Analysis

Codon usage bias is a significant characteristic of plant cp genomes and is typically influenced by factors such as genetic mutation and natural selection. The results revealed that the ENC plot, neutrality plot, and PR2 plot analyses yielded highly similar patterns for *M. guangnanica*, *M. hookeri*, and *M. longirostrata* (Figure 4). In the ENC plot analysis, the majority of genes in all three *Manglietia* cp genomes were below the standard curve. In the neutrality-plot analysis, the absolute values of the regression slopes were all less than 3%, with an R^2^ value of 0.022. These findings indicate that natural selection has a significant influence on the codon usage preferences of the three species. Furthermore, the PR2 plot analysis revealed that most genes in the cp genomes of all three *Manglietia* species fell within the fourth quadrant. This finding indicates that at the third codon position, uracil (U) is used more frequently than adenine (A), and guanine (G) is used more frequently than cytosine (C).

Relative synonymous codon usage (RSCU) is a key metric for quantifying the usage preference of synonymous codons. An RSCU value > 1 indicates that a codon is preferentially used, RSCU < 1 signifies that the codon is used less frequently than its synonymous alternatives, and RSCU = 1 denotes no usage bias [39]. RSCU analysis revealed that the highest RSCU values in *M. guangnanica*, *M.hookeri*, and *M. longirostrata* were observed for GCU (Ala) (1.865, 1.865, and 1.855), followed by AGA (Arg) (1.808, 1.808, and 1.801), and the lowest RSCU values were found for AGC (Ser) (0.294, 0.294, and 0.299) (Figure 5, Appendix A). The most frequently used codon across all three species was AUU (Ile), which occurred 754 times in the three species. This observation indicates that the cp genomes of these three species exhibit a high degree of conservation in terms of codon usage bias.

### 3.4. Comparative Genomic Analysis

JLB, JSB, JSA, and JLA denote the junction regions bordering LSC/IRb, SSC/IRb, SSC/IRa, and LSC/IRa, respectively, in the cp genome. Analysis of the IR boundary structures of the six *Manglietia* species revealed that the gene composition at the four junction regions (JLB, JSB, JSA, and JLA) is highly conserved. However, the IR regions exhibited varying degrees of expansion and contraction among the species. At the JLB boundary, the *rpl2* gene in the IRb region is located 60–61 bp from the boundary, whereas the *rps19* gene in the LSC region starts at the boundary. At the JSB boundary, the *ndhF* gene in the SSC region was located 60–73 bp from the boundary. The JSA boundary is crossed by the *ycf1* gene, with a length of 1279 bp in the IRa region and 4298–4352 bp in the SSC region. At the JLA boundary, except for the *trnH* gene in *M. decidua*, which is 88 bp from the boundary, the *trnH* genes in other species are 11–12 bp from the boundary (Figure 6).

MAUVE was used to align the chloroplast whole-genome sequences of the six *Manglietia* species, and the results revealed that the genomic sequences of all the species were highly conserved, with no instances of rearrangement or inversion detected (Figure 7). The mVISTA sequence alignment results demonstrated that the gene sequences within the cp genomes of the six *Manglietia* species are highly conserved and strongly consistent in size and orientation. As shown in Figure 8, noncoding regions exhibit greater variation than coding regions do. Among the coding regions, only the *ccsA* and *ycf1* genes presented more noticeable differences.

### 3.5. Selection Pressure Analysis

Comparing the rates of synonymous and nonsynonymous nucleotide substitutions is a method for assessing the selective pressure driving protein evolution. Ka represents the nonsynonymous substitution rate per nonsynonymous site, whereas Ks represents the synonymous substitution rate per synonymous site. Ka/Ks > 1 indicates that the gene is under positive selection, meaning that natural selection drives changes in the protein encoded by the gene; Ka/Ks = 1 indicates neutral evolution of the gene, meaning that the changes in the gene are neither beneficial nor harmful but rather the result of random genetic drift; Ka/Ks < 1 indicates that the gene is under purifying selection [40]. This study used the MLWL model of KaKs_calculator to calculate the Ka/Ks values among six *Manglietia* species. The results demonstrate that their protein-coding genes are highly conserved, with Ka/Ks values ranging from 0 to 1.5465, and that the majority of genes have Ka/Ks values < 1. When comparing *M. longirostrata* with *M.hookeri*, five genes had Ka/Ks > 0, whereas in comparison with *M. calcarea,* only the *ycf1* gene had a Ka/Ks > 0. These findings indicate that the protein-coding sequences of *M. longirostrata* and *M. hookeri* differ more significantly from those of *M. calcarea*. In the analysis of selective pressure between *M. guangnanica* and other *Manglietia* species, six genes had Ka/Ks > 0 compared with *M. calcarea*, whereas only *rpoA* had a Ka/Ks > 0 compared with *M.hookeri*. These findings indicate that the protein-coding sequences of *M. guangnanica* and *M. calcarea* differ more strongly from those of *M.hookeri*. In the comparison between *M. caveana* and *M. decidua*, eight genes had Ka/Ks > 0, with *ndhF*, *psaJ*, and *ycf1* values all exceeding 1. Compared with the other four *Manglietia* species, only *psaJ* and *rpoC1* presented Ka/Ks > 1. These findings suggest that these genes have undergone positive selection during evolution. Notably, except for the *ycf1* gene in the comparison between *M. longirostrata* and *M. hookeri*, where Ka/Ks = 0, the Ka/Ks values are nonzero in all the other *Manglietia* comparisons (Figure 9). This finding indicates that the *ycf1* gene has undergone varying degrees of nonsynonymous and synonymous substitutions in the bases across several *Manglietia* species.

### 3.6. Nucleotide Diversity and Highly Variable Regions

Nucleotide diversity (Pi) values can be used to assess genetic variation among species. Regions with high Pi values are often suitable for analyzing genetic diversity among samples at familial, generic, or lower taxonomic levels [41]. This study analyzed nucleotide diversity in 31 *Manglietia* cp genomes to investigate variation. The Pi values ranged from 0 to 0.01251, with an average of 0.00103. Among these, the Pi values in the LSC region ranged from 0 to 0.01251, with an average of 0.00122; the Pi values in the SSC region ranged from 0.0001 to 0.00859, with an average of 0.00211; and the Pi values in the IR region ranged from 0 to 0.00196, with an average of 0.00030. Figure 10 shows the eight highly variable regions on the cp genome with Pi values > 0.005, including five intergenic regions (*petA-psbJ*, *rpl32-trnL*, *trnT-trnL*, *rps15-ycf1*, and *ccsA-ndhD*) and three genes (*psbJ*, *ccsA*, and *ndhD*). Among these regions, the *petA-psbJ* region has the highest Pi value (0.1088), and all these highly variable regions are located in the LSC or SSC regions, while the IR region is more conserved than the single-copy region.

### 3.7. Phylogenetic Analysis

To further clarify the phylogenetic positions of *M. guangnanica*, *M. longirostrata*, and *M.hookeri* within *Manglietia* and resolve infrageneric relationships, this study incorporated the cp genomes of two *Houpoea* species in addition to the 31 *Manglietia* genomes. Using the two *Houpoea* species as the outgroup, phylogenetic trees were constructed via both Maximum Likelihood (ML) and Bayesian inference methods. The ML tree and MrBayes tree constructed on the basis of 85 protein-coding sequences from the cp genomes revealed that *Manglietia* comprises four primary clades: Clade I, Clade II, Clade III, and Clade IV. Clade I includes 18 *Manglietia* species; Clade II includes only *M. decidua*; Clade III includes 9 *Manglietia* species; Clade IV includes only *M. caveana* J. D. Hooker & Thomson. Among these, *M. longirostrata* is classified into Clade I and clusters with *M. dandyi* (Gagnepain) Dandy and *M. Zhengyiana* N. H. Xia. *M. hookeri* and *M. guangnanica* are included in Clade II and cluster with *M. grandis* Hu & W. C. Cheng and *M. crassipes*. These nodes exhibit high bootstrap support and posterior probability in the phylogenetic tree (Figure 11).

## 4. Discussion

### 4.1. Chloroplast Genome Characteristics of Three Manglietia Species

The cp genomes of angiosperms exhibit conserved structures and sizes, typically consisting of two single-copy regions (LSC and SSC) and two identical inverted repeat regions (IRb and IRa), with sizes predominantly ranging from 120 to 160 kb [42]. The three newly sequenced *Manglietia* cp genomes in this study also display this typical quadripartite structure. The lengths of these genes range from 160,067 to 160,076 bp, each containing 133 annotated genes, and all share an identical GC content of 39.3%. These characteristics align closely with those previously reported for other Magnoliaceae species [2,6,14]. The GC content of the IR regions (43.2%) of all three *Manglietia* species was significantly greater than that of the LSC region (38%) and SSC region (34.2%). This phenomenon is widespread among other plant groups, such as *Saxifraga* (42.8%) [43] and *Impatiens* (43%) [44], and is attributed primarily to the presence of four highly conserved rRNA genes with elevated GC contents located within the IR regions. Recombination can occur between similar sequences, so repetitive sequences are often considered major drivers of genetic variation in genomes [45]. This study revealed 31, 30, and 30 dispersed repeats in the cp genomes of *M. guangnanica*, *M. hookeri*, and *M. longirostrata*, respectively. Only forward repeats (F) and palindromic repeats (P) were identified, a characteristic that is consistent with previous reports for other *Manglietia* species [14]. In *M. guangnanica* and *M. hookeri*, the number of palindromic repeats exceeds that of forward repeats, with the former having an additional 75 bp forward repeat compared with the latter. *M. longirostrata* differs significantly from the other two, as its cp genome contains more forward repeats than palindromic repeats, with a more diverse range of lengths. Simple sequence repeats (SSRs), characterized by their high abundance and wide distribution in genomes, serve as efficient molecular markers for species identification and studies of genetic diversity [46,47]. In this study, 53, 53, and 56 SSRs were detected in the cp genomes of *M. guangnanica*, *M. hookeri*, and *M. longirostrata*, respectively. The number of SSRs differed slightly among the three species. With respect to SSR type, mononucleotide repeats composed of A/T units predominated in all three groups, and no pentanucleotide repeats were detected. This characteristic is common in other Magnoliaceae species [2,6]. In terms of distribution, these SSRs are not evenly distributed across the genome. The LSC region contains more SSRs than the SSC and IR regions do, and noncoding regions contain more SSRs than coding regions do. This phenomenon has also been observed in the cp genomes of other plant families [48,49].

The effective number of codons (ENC) can be used to quantify the preference of genes for using codons, with a range from 20 to 61, typically with 35 as the standard. A higher value indicates more balanced codon usage [50]. Neutrality-plot analysis assesses the factors influencing codon usage bias by examining the correlation between nucleotide composition at codon positions 1 and 2 (GC12) and position 3 (GC3). If GC12 and GC3 are significantly correlated and the regression slope is close to 1, this indicates that codon usage bias is influenced primarily by mutation pressure. Conversely, if not, natural selection is the dominant factor [51]. PR2-plot analysis primarily examines the proportional relationships among the four nucleotides at the third codon position across genes in the genome. If the four bases are uniformly distributed across the plot regions, mutation pressure may dominate codon usage bias; otherwise, natural selection is the main influencing factor [52]. The results of the ENC plot, neutrality plot, and PR2 plot analyses of the cp genomes of *M. guangnanica*, *M. hookeri*, and *M. longirostrata* consistently indicate that natural selection is the dominant factor influencing their codon usage bias. The relative synonymous codon usage (RSCU) value is primarily used to assess the degree of codon usage bias. RSCU patterns often vary among species from different families or genera [51]. In this study, the RSCU values of the cp genomes of *M. guangnanica*, *M. hookeri*, and *M. longirostrata* were extremely similar, indicating that their protein-coding sequences are highly conserved in codon selection preferences. Consistent with those of other angiosperms, the cp genomes of *M. guangnanica*, *M. hookeri*, and *M. longirostrata* also demonstrated a preference for codons ending in A/U.

Genomic regions with high nucleotide diversity have been widely utilized as molecular markers for the identification and phylogenetic analysis of different plant groups [53]. In this study, the evaluation of nucleotide diversity across 31 *Manglietia* cp genomes identified eight highly variable regions. Notably, three of these regions (*petA–psbJ, rpl32–trnL, and ccsA–ndhD*) have also been reported in *Houpoea* and *Lirianthe*, which belong to Magnoliaceae [2,6]. On the basis of the nucleotide diversity analysis conducted in this study, we recommend *petA-psbJ*, *rpl32-trnL*, and *ccsA-ndhD* as effective molecular markers for species identification and phylogenetic studies within *Manglietia*. Highly variable regions are often hotspots for mutational events such as insertions or deletions (InDels) and single-nucleotide polymorphisms (SNPs). These events contributed to their accelerated evolutionary rate [54]. Understanding and applying knowledge of these hypervariable genomic regions will enhance species identification and conservation efforts within *Manglietia*.

### 4.2. Interspecific Divergence of Chloroplast Genomes

To clarify the interspecific relationships between *M. longirostrata* and *M.hookeri*, as well as between *M. guangnanica* and *M. calcarea*, and to further investigate the divergence among the cp genomes of the *Manglietia* species, structural comparisons and selective pressure analyses were conducted on the cp genomes of the aforementioned four species, along with those of *M. caveana* and *M. decidua*. The cp genomes of higher plants are highly conserved. The expansion and contraction of inverted repeat (IR) regions significantly influence variations in cp genome size and often lead to the formation of pseudogenes [55]. In this study, the IR boundaries of the six *Manglietia* cp genomes exhibited a high degree of conservation, and the genes located at the four junction sites (JLB, JSB, JSA, and JLA) were identical across species. However, the extent of IR expansion and contraction slightly varied. In terms of IR expansion and contraction, *M. decidua* differs significantly from the other five *Manglietia* species; *M. longirostrata* is similar to *M. calcarea*, and *M. hookeri* is similar to *M. guangnanica*. Mauve alignment analysis revealed that the six *Manglietia* species not only presented highly conserved genome structures but also maintained strong consistency in terms of gene sequence size and orientation. The observed divergence was derived primarily from noncoding regions of the genome. Coding regions, which contain numerous protein-coding genes, are typically more conserved than noncoding regions. In the mVISTA analysis, six regions (*trnT-trnL*, *petA-psbJ*, *rpl32-trnL*, *psbJ-psbL*, *ccsA*, and *ycf1*) were significantly different across the cp genomes of the six *Manglietia* species. The Ka/Ks ratio is widely employed to measure selective pressure on genes and is associated with adaptive evolution [56]. In most chloroplast genes, synonymous substitutions are more frequently observed than nonsynonymous substitutions are, so Ka/Ks values are typically less than 1 [57]. This study revealed that the Ka/Ks ratios for protein-coding genes among *M. longirostrata*, *M.hookeri*, *M. guangnanica*, and *M. calcarea* were all less than 1, indicating that their protein-coding genes are under purifying selection. Ka/Ks analysis further revealed a closer phylogenetic relationship between *M. longirostrata* and *M. calcarea*, while *M. guangnanica* had a stronger affinity for *M. hookeri*. Furthermore, comparisons involving *M. caveana* revealed that the *psaJ* and *rpoC1* genes presented Ka/Ks values > 1 compared with those of the aforementioned four *Manglietia* species, and the *ndhF*, *psaJ*, and *ycf1* genes presented Ka/Ks values > 1 compared *with those of M. decidua*, indicating that these genes are under positive selection. Nonsynonymous substitutions (Ka) can induce amino acid changes, driving functional alterations to adapt to environmental pressures [58]. Functionally, *psaJ* encodes a subunit of photosystem I (PS I) [59], and *ndhF* encodes a subunit of the chloroplast NADH dehydrogenase-like complex (NDH) involved in photosynthetic electron transport [60], both of which play crucial roles in photosynthetic reactions. *rpoC1* encodes the β’ subunit of RNA polymerase, a key subunit in the transcription process [61]. *ycf1* is one of the longest genes in the plant cp genome and is located at the boundary between the IR region and the SSC region. It is often affected by the expansion or contraction of the IR region, becomes a pseudogene, and serves as a valuable marker for phylogenetics and species identification [62,63]. In the cp genomes of the six *Manglietia* species, these genes presented Ka/Ks > 1, indicating that they are undergoing rapid evolution. Further study of these genes is crucial for elucidating the adaptive evolution of *Manglietia* species.

### 4.3. Phylogenetic Analysis

Owing to their uniparental inheritance and smaller structure and more moderate evolutionary rate than mitochondrial or nuclear genomes do, cp genomes have demonstrated considerable utility in resolving phylogenetic relationships. This study utilized 85 protein-coding genes from the cp genomes of 31 *Manglietia* species and two *Houpoea* species to construct both maximum likelihood (ML) and MrBayes phylogenetic trees. Owing to the incorporation of a significantly larger number of *Manglietia* cp genomes, our results differ from those reported by Li et al. [14]. In our analysis, *Manglietia* was resolved into four distinct clades: clade I, clade II, clade III, and clade IV. Clade I included 18 *Manglietia* species, Clade II included only *M. decidua*, Clade III included 9 *Manglietia* species, and Clade IV included only *M. caveana*. The infrageneric classification systems for *Manglietia* proposed by previous studies on the basis of morphology were not supported by our study. The traditionally defined Sect. *Manglietia* was found not to be monophyletic, with many of its constituent species distributed across different clades in our study (Figure 12). However, Sect. Decidua proposed by Zheng [12] and Sima [9], which contains only *M. decidua*, is supported by our phylogenetic results. Among the infrageneric classification systems proposed by Xia [13] and Sima [9], *M. caveana* is considered to be closely related to *M. szechuanica* Hu, *M. dulclouxii* Finet & Gagnepain, and *M. ventii* N. V. Tiep because of its unique characteristic of a pubescent gynoecium and is therefore classified under the subgenus *Manglietia* or subsection *Manglietia*. However, in this study, *M. caveana* was classified into a separate clade and located at the base of the phylogenetic tree. M. caveana is primarily distributed in Motuo County, XiZang Autonomous Region, China [8]. Motuo County is located on the southern slope of the eastern end of the Himalayas and is surrounded by high mountains on three sides, with significant vertical elevation differences. Owing to the influence of warm and moist air currents from the Indian Ocean and the barrier effect of the surrounding mountains, the region experiences extremely abundant rainfall and a complete vertical climate zone, encompassing nearly all climate zones of the Northern Hemisphere [64]. To adapt to these unique geographical and climatic conditions, adaptive evolution may have occurred in specific genes within the *M. caveana* genome (such as *psaJ*, *rpoC1*, *ndhF*, and *ycf1*), potentially leading to genetic divergence from other *Manglietia* species. Both Clades I and III in our study encompass multiple *Manglietia* species. Further morphological examination revealed that the majority of the species in Clade I presented pubescence on twigs, stipules, and the leaf surface, whereas the species in Clade III were typically glabrous or became glabrous and grew in these structures. The presence or absence of pubescents on twigs, stipules, and leaf surfaces has shown potential in elucidating the infrageneric classification system within *Manglietia*. On the basis of herbarium specimen observations, Hu et al. suggested that *M. guangnanica* shares close morphological similarities with *M. pachyphylla*, *M. crassipes*, and *M.fordiana*, implying a close phylogenetic relationship [18]. In this study, *M. guangnanica* was classified into Clade III, which clustered with *M. crassipes* and presented high support (BS = 94, PP = 0.99). Moreover, *M. pachyphylla* and *M. fordiana* were both classified into Clade I. These findings indicate that *M. guangnanica* shares a relatively close phylogenetic relationship with only *M. crassipes*. Furthermore, *M. calcarea* was also assigned to Clade I, confirming its distant relationship with *M. guangnanica*. Morphologically, *M. guangnanica* and *M. calcarea* are not closely related. Their primary commonality is their distribution in mixed forests of limestone areas in southwestern China, where their populations often intermingle. This sympatric distribution may be the main reason for the confusion between their species identities. The infrageneric taxonomic status of *M. longirostrata* has been contentious. Hu et al. treated it as a variety of *M.hookeri* (*M.hookeri* var. *longirostrata*) [15], whereas Sima et al. argued for its recognition as a distinct species within *Manglietia* [16]. In our phylogenetic analysis, *M. longirostrata* was placed in Clade I, clustering with *M. dandyi* and *M. zhengyiana* (BS = 57, PP = 0.76), whereas *M. hookeri* was located in Clade III. Morphologically, compared with *M. hookeri*, *M. longirostrata* has a longer gynoecium and aggregate fruits, smaller carpels, glabrous petioles, and glabrous and longer peduncles. Additionally, there are differences in the pubescence of buds and twigs. In combination with the comparative results of the cp genomes, this study concludes that classifying *M. longirostrata* as a variety of *M.hookeri* is taxonomically unsupported.

## 5. Conclusions

This study sequenced, assembled, and annotated the cp genomes of *Manglietia guangnanica*, *Manglietia hookeri*, and *Manglietia longirostrata*. The results revealed highly conserved cp genome structures and sizes across these species. We detected 53, 53, and 56 SSRs, respectively, which serve as effective molecular markers for investigating genetic diversity among their populations. Analyses of the ENC plot, neutrality plot, and PR2 plot indicated that natural selection is the dominant factor influencing codon usage bias in their cp genomes. Analyses of IR boundaries, synteny, sequence alignment, and selection pressure in six *Manglietia* species revealed that their cp genomes exhibit variability in length and structure, with most variation concentrated in the noncoding regions of the cp genome. Furthermore, nucleotide diversity analysis across 31 *Manglietia* species revealed eight highly variable regions, three of which (*petA-psbJ*, *rpl32-trnL*, and *ccsA-ndhD*) were selected as prime candidate molecular markers. These regions provide valuable genomic resources for species identification and phylogenetic studies within *Manglietia*. The phylogenetic analysis in this study divided *Manglietia* into four clades, which differs from the previously proposed infrageneric classification systems. In terms of interspecific relationships, *M. longirostrata* and *M. hookeri* were placed in different clades, while *M. guangnanica* and *M. calcarea* were also placed in different clades, both with high support values (PP = 75, BS = 1). Furthermore, based on chloroplast genome characteristics and protein coding sequence similarity, *M. longirostrata* is more similar to *M. calcarea*, while *M. hookeri* is similar to *M. guangnanica*. Therefore, this study considers *M. guangnanica* and *M. calcarea* to be two distinct species, while the treatment of *M. longirostrata* as a variety of *M. hookeri* is not supported.

## Figures and Tables

**Figure 1 cimb-47-00737-f001:**
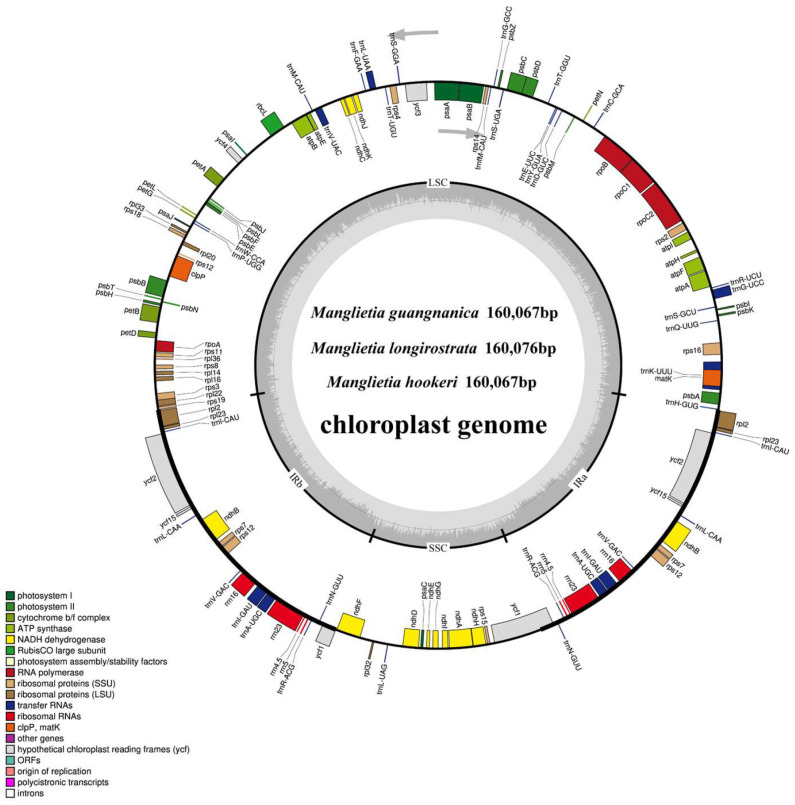
The cp genomes of the three *Manglietia* species. The circular map depicts a representative cp genome (*M. guangnanica*). In the inner circle, the darker gray represents the GC content, and the lighter gray represents the AT content. Genes inside the circle are transcribed clockwise, whereas those outside are transcribed counterclockwise. Different colors represent genes with different functional categories in the cp genome. The thick lines on the outer circle indicate the boundaries of the inverted repeat regions (IRa and IRb), whereas the remaining sections represent the boundaries of the single-copy (LSC and SSC) regions. IR, inverted repeats; LSC, large single-copy; SSC, small single-copy.

**Figure 2 cimb-47-00737-f002:**
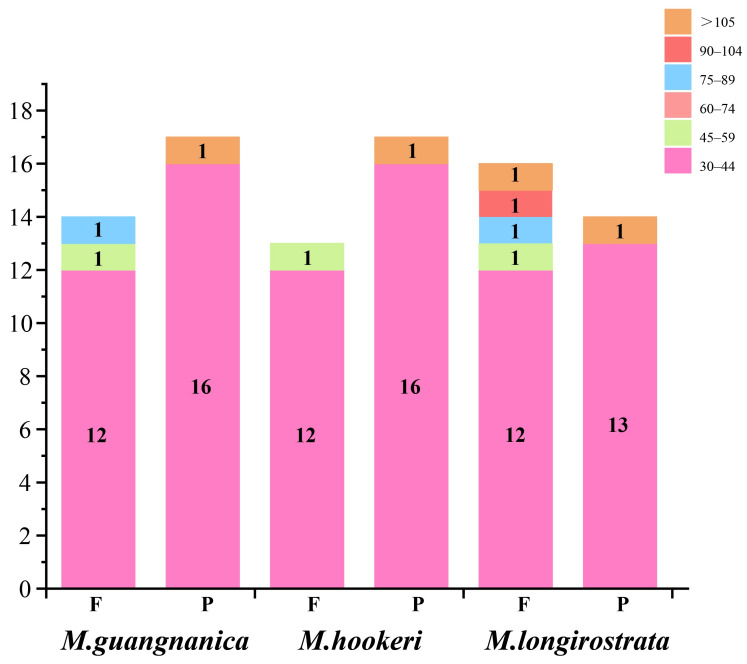
Repeat sequences in the cp genomes of three *Manglietia* species. The horizontal axis represents the type of repeat, and the vertical axis represents the number of repeats for each type. F: forward repeats; P: palindromic repeats. Different colors represent repeats of different length ranges (bp). The numbers in the columns indicate the number of repetitions.

**Figure 3 cimb-47-00737-f003:**
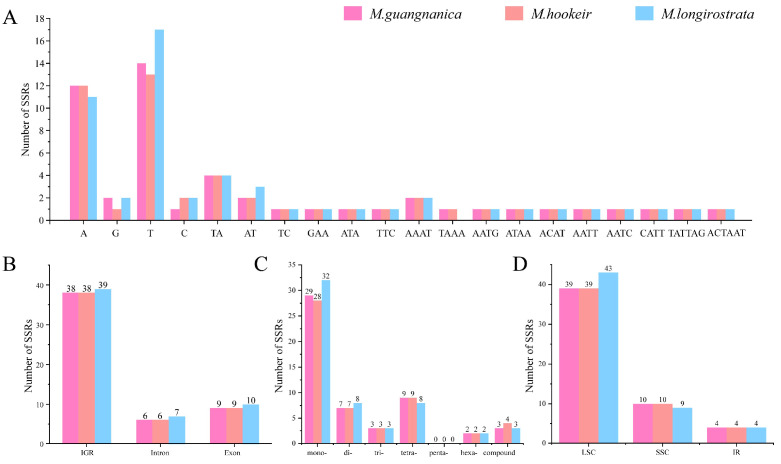
Simple sequence repeats (SSRs) in the cp genomes of the three *Manglietia* species. Different colors represent different species. (**A**) The X-axis shows statistics of SSR unit types, and the Y-axis shows the number of corresponding repeat units. (**B**) The X-axis shows the genomic locations of SSRs, and the Y-axis shows the number of SSRs at the corresponding locations. (**C**) The X-axis shows SSR types (based on motif size), and the Y-axis shows the number of SSRs for each corresponding repeat type. (**D**) The X-axis shows regions of the cp genome (LSC, SSC, and IR), and the Y-axis shows the number of SSRs in the corresponding region.

**Figure 4 cimb-47-00737-f004:**
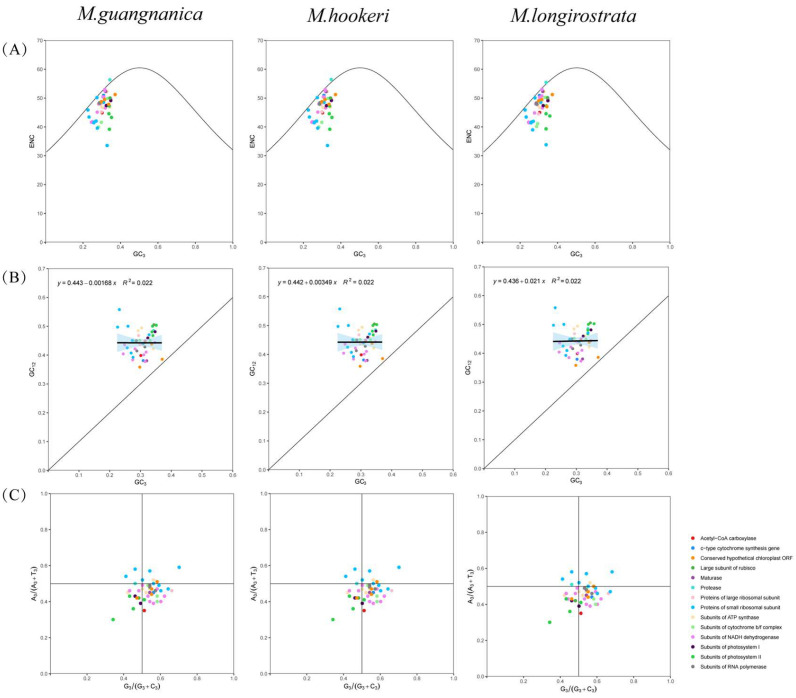
Codon Usage Bias Analysis in the Three *Manglietia* Species. (**A**) represents the ENC plot analyses; (**B**) represents neutrality plot analyses; (**C**) represents PR2 plot analyses. Points of different colors denote genes belonging to different functional categories. From left to right, the species shown are *M. guangnanica*, *M. hookeri*, and *M. longirostrata*.

**Figure 5 cimb-47-00737-f005:**
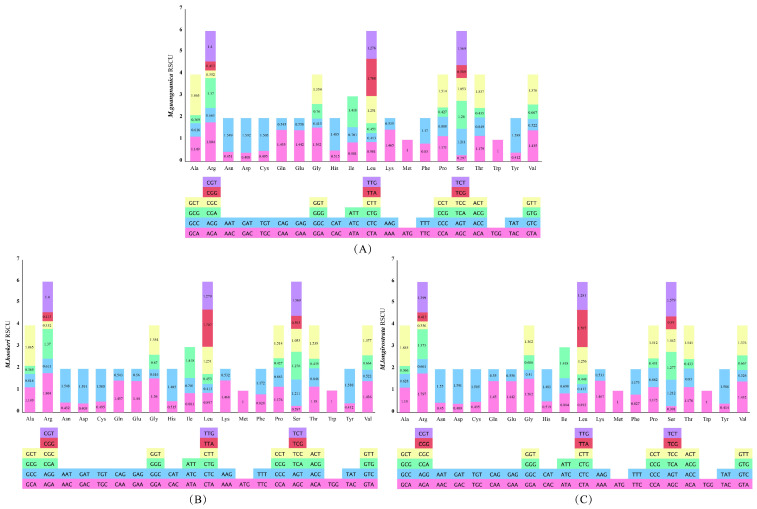
RSCU analysis of the three *Manglietia* species. (**A**), (**B**), and (**C**) represent the RSCU value statistics for *M. guangnanica*, *M. hookeri*, and *M. longirostrata*, respectively. For each panel, the X-axis denotes the amino acids, the Y-axis represents the RSCU values, and different colors correspond to different codons.

**Figure 6 cimb-47-00737-f006:**
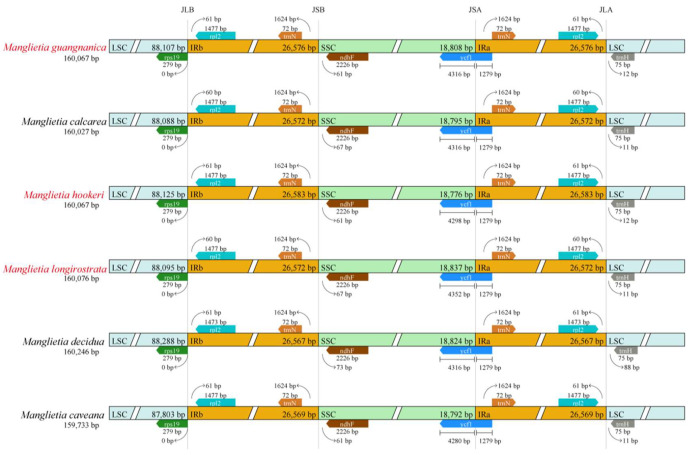
IR boundary analysis of six *Manglietia* species. JLB (IRb/LSC), JSB (IRb/SSC), JSA (SSC/IRa), and JLA (IRb/LSC) represent the junction sites between two adjacent regions in the cp genome.

**Figure 7 cimb-47-00737-f007:**
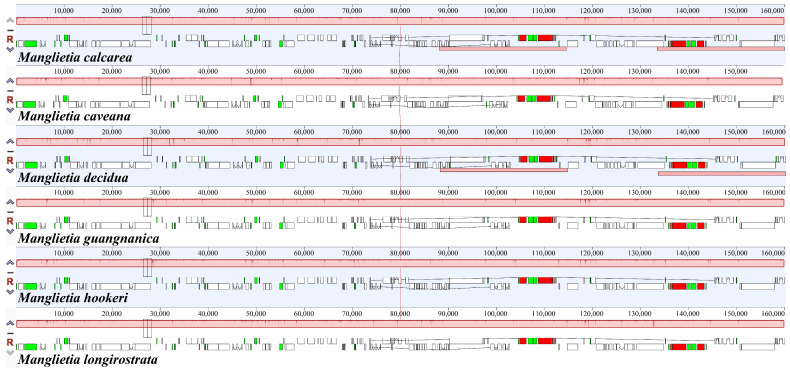
Mauve alignment of the cp genomes of six *Manglietia* species. The red vertical line connects homologous regions of sequences across different chloroplast genomes. White represents locally collinear blocks of CDS exons, green denotes locally collinear blocks of tRNA genes, and red indicates locally collinear blocks of rRNA genes.

**Figure 8 cimb-47-00737-f008:**
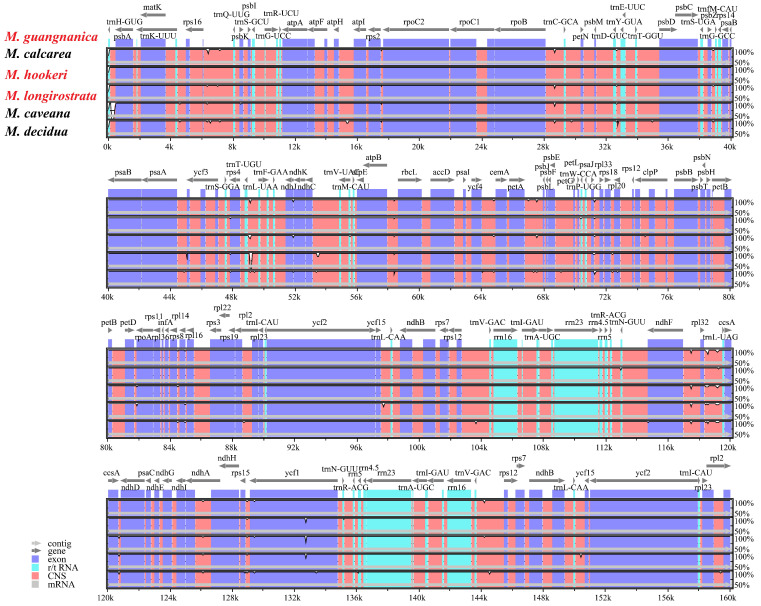
Comparison of the structures of the six cp genomes, with *M. guangnanica* used as a reference. The red text denotes the species sequenced in this study.The X-axis represents the position of the cp genome, and the Y-axis represents different species. Sequence similarity is indicated by 50–100%.

**Figure 9 cimb-47-00737-f009:**
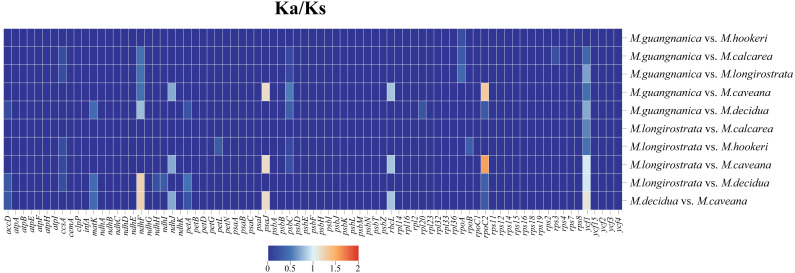
Ka/Ks Analysis of Protein-Coding Genes in the cp Genomes of Six *Manglietia* Species.

**Figure 10 cimb-47-00737-f010:**
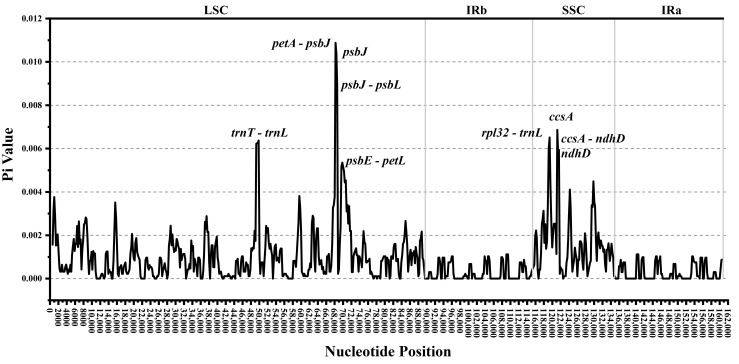
Comparative Analysis of Nucleotide Diversity (Pi) in the cp genomes of 31 *Manglietia* Species. The X-axis represents the position along the cp genome sequence. The Y-axis represents the nucleotide diversity (Pi) value.

**Figure 11 cimb-47-00737-f011:**
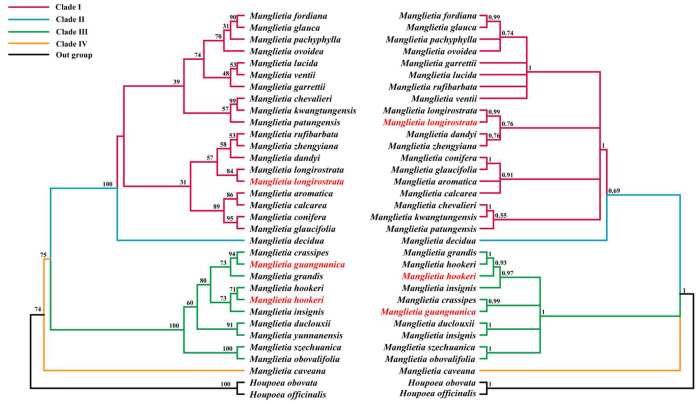
Phylogenetic Tree Constructed Based on 85 Protein-Coding Genes from the cp Genomes of 31 *Manglietia* Species and 2 *Houpoea* Species. The red text denotes the species sequenced in this study.The left panel shows the phylogenetic tree constructed via the maximum likelihood method with 1000 repetitions, with bootstrap support (BS) values displayed at the nodes. The right panel shows the phylogenetic tree of *Manglietia* constructed via the MrBayes method with 1,000,000 generations, with values displayed at the nodes representing posterior probability (PP). Unlike previous classifications, Clades I–IV represent branches within the genus *Manglietia* defined by this study.

**Figure 12 cimb-47-00737-f012:**
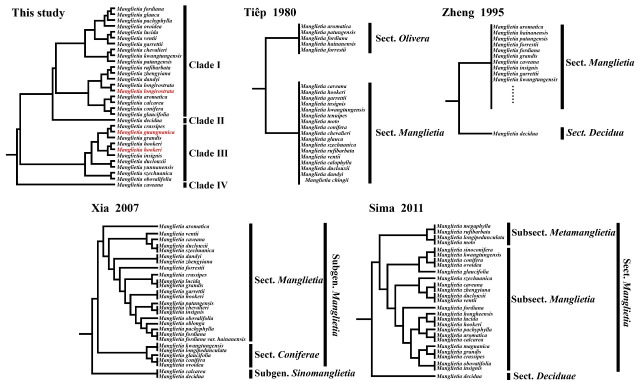
The different infrageneric classifications of *Manglietia.* The red text denotes the species sequenced in this study.

**Table 1 cimb-47-00737-t001:** Characteristics of the complete cp genomes of the three *Manglietia* species.

Species	Total Length (bp)	Total GC (%)	LSC Length (bp)	LCS GC (%)	SSC Length (bp)	SSC GC (%)	IR Length (bp)	IR GC (%)	Genes	CDS	tRNAs	rRNAs	Number of Accession
*M.guangnanica*	160,067	39.3	88,107	38	18,808	34.2	26,576	43.2	133	88	37	8	PQ246290
*M.longirostrata*	160,076	39.3	88,095	38	18,837	34.2	26,572	43.2	133	88	37	8	PV239601
*M.hookeri*	160,067	39.3	88,125	38	18,776	34.2	26,583	43.2	133	88	37	8	PV239602

Note: GC, guanine–cytosine; CDS, protein-coding gene.

**Table 2 cimb-47-00737-t002:** Gene information annotated in the cp genomes of three Manglietia species.

Category	Gene
Large ribosomal subunits	*rpl2*(2) *, *rpl14*, *rpl16*, *rpl20*, *rpl22*, *rpl23*(2), *rpl32*, *rpl33*, *rpl36*
Small ribosomal subunits	*rps2*, *rps3*, *rps4*, *rps7*(2), *rps8*, *rps11*, *rps12*(2) **, *rps14*, *rps15*, *rps16* *, *rps18*, *rps19*
DNA-dependent RNA Polymerase protease	*rpoA*, *rpoB*, *rpoC1* *, *rpoC2*
Translation initiation factor	*infA*
Ribosomal RNA genes	*rrn4.5*(2), *rrn5*(2), *rrn16*(2), *rrn23*(2)
Transfer RNA genes	*trnA-UGC*(2) *, *trnC-GCA*, *trnD-GUC*, *trnE-UUC*, *trnF-GAA*, *trnfM-CAU*, *trnG-GCC*, *trnG-UCC* *, *trnH-GUG*, *trnI-CAU*(2), *trnI-GAU*(2) *, *trnK-UUU* *, *trnL-CAA*(2), *trnL-UAA* *, *trnL-UAG*, *trnM-CAU*, *trnN-GUU*(2), *trnP-UGG*, *trnQ-UUG*, *trnR-ACG*(2), *trnR-UCU*, *trnS-GCU*, *trnS-GGA*, *trnS-UGA*, *trnT-GGU*, *trnT-UGU*, *trnV-GAC*(2), *trnV-UAC* *, *trnW-CCA*, *trnY-GUA*
Photosystem I Photosystem II Rubisco large subunit Cytochrome b6/f complex F-type ATP synthase DNA-dependent RNA Polymerase protease NAD(P)H dehydrogenase complex Maturase Inner membrane protein Subunit of acetyl-CoA-carboxylase Cytochrome C biogenesis protein Function uncertain	*psaA*, *psaB*, *psaC*, *psaI*, *psaJ* *psbA*, *psbB*, *psbC*, *psbD*, *psbE*, *psbF*, *psbH*, *psbI*, *psbJ*, *psbK*, *psbL*, *psbM*, *psbN*, *psbT*, *psbZ* *rbcL* *petA*, *petB* *, *petD*, *petG*, *petL*, *petN* *atpA*, *atpB*, *atpE*, *atpF* *, *atpH*, *atpI* *clpP* ** *ndhA* *, *ndhB*(2) *, *ndhC*, *ndhD*, *ndhE*, *ndhF*, *ndhG*, *ndhH*, *ndhI*, *ndhJ*, *ndhK* *matK* *cemA* *accD* *ccsA* *ycf1*(2), *ycf2*(2), *ycf3* **, *ycf4*, *ycf15*(2)

Note: Gene (2): the number of multicopy genes; Gene *: one-intron-containing gene; Gene **: two-intron gene.

## Data Availability

The chloroplast genome sequences for the species studied are publicly available at the National Center for Biotechnology Information (NCBI) under the following accession numbers: *Manglietia guangnanica* (PQ246290), *Manglietia hookeri* (PV239602), and *Manglietia longirostrata* (PV239601).

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
