# Peer review of "The Complete Chloroplast Genomes of Three Manglietia Species and Phylogenetic Insight into the Genus Manglietia Blume"

_cimb, 2025, doi:10.3390/cimb47090737_

Round 1

Reviewer 1 Report

Comments and Suggestions for Authors

representatives of the Manglietia family, which are an economically important group of plants, which is also of great importance in ornamental horticulture. The classifications for representatives of the Manglietia family currently available to researchers are mainly based on morphological differences. However, weak differences in the morphological features used for many species of this family complicate intraspecific classification, as well as the analysis of interspecific relationships. In this regard, the search for new, more reliable molecular markers using chloroplast genome sequencing data seems to be a relevant and important task for the classification of plant species and the assessment of their genetic diversity during evolutionary development. The authors present experimental data on the analysis of the nucleotide composition of chloroplast genomes of three species of the Manglietia family: M. guangnanica, M. hookeri and M longirostrata, which are clearly displayed in two tables and ten figures. To perform sequencing and annotation of the obtained data, generally accepted methods of molecular and bioinformatics data analysis were used. Using the genome sequencing data of the three studied species, the authors performed a phylogenetic analysis, the results of which are presented in Figure 11, and made important clarifying conclusions about the intrageneric classification of the genus Manglietia. The manuscript is well structured, the cited sources of literature are relevant, more than half of them relate to the works of the last five years. The data presented by the authors are well substantiated and can be used in further evolutionary studies of the genus Manglietia. This manuscript can be recommended for publication in the journal in the presented form.

Author Response

Comments and Suggestions for Authors

representatives of the Manglietia family, which are an economically important group of plants, which is also of great importance in ornamental horticulture. The classifications for representatives of the Manglietia family currently available to researchers are mainly based on morphological differences. However, weak differences in the morphological features used for many species of this family complicate intraspecific classification, as well as the analysis of interspecific relationships. In this regard, the search for new, more reliable molecular markers using chloroplast genome sequencing data seems to be a relevant and important task for the classification of plant species and the assessment of their genetic diversity during evolutionary development. The authors present experimental data on the analysis of the nucleotide composition of chloroplast genomes of three species of the Manglietia family: M. guangnanica, M. hookeri and M longirostrata, which are clearly displayed in two tables and ten figures. To perform sequencing and annotation of the obtained data, generally accepted methods of molecular and bioinformatics data analysis were used. Using the genome sequencing data of the three studied species, the authors performed a phylogenetic analysis, the results of which are presented in Figure 11, and made important clarifying conclusions about the intrageneric classification of the genus Manglietia. The manuscript is well structured, the cited sources of literature are relevant, more than half of them relate to the works of the last five years. The data presented by the authors are well substantiated and can be used in further evolutionary studies of the genus Manglietia. This manuscript can be recommended for publication in the journal in the presented form.

Response: Thank you so much.We greatly appreciate your professional review and recognition of our article. The article ‘The Complete Chloroplast Genome of Three Manglietia Species and Phylogenetic Insight into the Genus Manglietia Blume’ by Luo et al. The aim was to clarify Classification system and interspecific relationships within the Manglietia genus. In this study, the authors classified 28 Manglietia species into 4 clades based on chloroplast genomes and clarify the interspecific relationships between M. longirostrata and M.hookeri, as well as between M. guangnanica and M. calcarea. This enriches the cp genome data of Manglietia, provides new insights into infrageneric classification, and lays a foundation for further phylogenetic and evolutionary studies of Manglietia.

Reviewer 2 Report

Comments and Suggestions for Authors

I have reviewed your article and will provide comments aimed at making minor corrections to help improve the quality of the manuscript.

The Complete Chloroplast Genome of Three Manglietia Species and Phylogenetic Insight into the Genus Manglietia Blume

Yuan Luo, Wei Luo, Tongxing Zhao, Jing Yang, Lang Yuan, Pinzheng Zhang, Zixin Gong, Haizhu Li,Yongkang Sima

The authors present the sequencing and analysis of chloroplast genomes in Manglietia spp. Although there are several studies related to genome sequencing in the Manglieta genus, I find this work particularly interesting for the field. I believe this work can serve as a basis for future genetic and evolutionary studies related to the species. I have reviewed your article and will provide comments aimed at making minor corrections to help improve the quality of the manuscript.

Title: Overall, the title of the manuscript is clear. However, since the study involves three species, it would be more appropriate to use the plural form. Regarding nomenclature, please ensure that the genus Manglietia is italicized throughout the document, in accordance with standard taxonomic conventions. Suggested revised title: “The Complete Chloroplast Genomes of Three Manglietia Species and Phylogenetic Insights into the Genus Manglietia Blume”

Line 43: It is recommended to define the abbreviation at its first mention, for example: “the chloroplast (cp) genome.”

Line 48: Please use the abbreviation “cp genome” after the first mention of “chloroplast genome.”

Line 53: The authors should use abbreviations after the first full mention. Please review and apply this throughout the manuscript (e.g., to increasingly favor the use of cp genomes).

Line 72: The full name of the genus Manglietia is appropriately introduced at the beginning. To avoid redundancies and improve readability, you may use the abbreviated form of the genus, as long as it does not cause ambiguity. (e.g., M. fordiana).

Line 131-132: The authors should use a consistent font style and size throughout all sections, including references to databases and URLs.

Line 152: It is no longer necessary to mention the full genus name, even when referring to a different species. Please review and apply this throughout the manuscript (e.g., The cp genomes of M. guangnanica, M. longirostrata, and M. hookeri).

Line 155: There appears to be an error in the reference management. Authors should review and update cross-references to ensure they are correctly linked, especially to figures and tables.

Line 168: The figure caption lacks sufficient detail to fully describe the results. While it may seem redundant, a complete and accurate figure description is necessary for clarity and proper interpretation.

Line 175: Table1. The column headers in the table are not centered. Please adjust the alignment to ensure consistency and improve the table's readability.

Line 191-192: There appears to be the same reference error mentioned above, likely caused by the reference manager. It seems the authors intended to refer to Figure 2, but the current placeholder "(Error! Reference source not found.)" Is confusing and disrupts the flow of the text.

Line 194: The figure caption is too general. While it may seem redundant, a complete and accurate figure description is necessary for clarity and proper interpretation.

Line 356: Based on the objective analysis, what is the biological or evolutionary significance of the distribution of Manglieta across the different clade? 

Line 532: I recommend repositioning either the figure or its initial reference to improve clarity, as its current placement gives the impression that it is disconnected from the content. If the figure is not essential to the main text, consider moving it to the supplementary materials.

Line 542: The authors should indicate how the integrity of the DNA was assessed (e.g., agarose gel electrophoresis).

Line 546: The subtitle on line 542 is repeated.

Line 547: The authors should include additional information such as the type of library, read length, amount of data generated, and the parameters used during data filtering.

Line 548: It is important to mention the parameters used for genome assembly or whether default settings were applied. Also indicate the total number of reads used or the size of the input file.

Line 554: What version of OGDRAW did you use to generate the chloroplast genome maps?

Line 555: Preferably add the software versions please.

Line 567: The authors should use a consistent font style and size.

Line 571: For EMBOSS the authors do not specify the version, it would be useful to add it if known.

Line 571-572: Briefly explain what these analyses measure.

Line 589: “KaKs_Calculator” is mentioned, but neither the version nor the source (link or source) are indicated. This is important for reproducibility.

Line 631: The authors make an important taxonomic claim: “this study considers M. guangnanica and M. calcarea to be two distinct species.” However, sufficient phylogenetic support values to substantiate this conclusion are not provided. Moreover, the level of genomic divergence between these taxa is not discussed.

Line 634: The supplementary tables mentioned were not found, except for Table S6. It is recommended to verify that all supplementary materials are properly included. The genus names of species should be written correctly and italicized.

Line 649: This should also be included in the Materials and Methods section.

Line 654: The reference list needs careful revision. First, the formatting is inconsistent, with a mixture of citation styles and font sizes. Please ensure that the reference formatting is standardized according to the journal’s guidelines and that DOIs are provided for all references where applicable. The DOI in references is recommended to be written in its complete form as a URL link: https://doi:10.1093/pcp/pcn069.

Line 801-line 804: The paragraph is repeated again. It is recommended to carefully review the manuscript to eliminate unnecessary repetitions and improve the clarity of the text.

Author Response

Response to Referee 2

The article ‘The Complete Chloroplast Genome of Three Manglietia Species and Phylogenetic Insight into the Genus Manglietia Blume’ by Luo et al. The aim was to clarify Classification system and interspecific relationships within the Manglietia genus. In this study, the authors classified 28 Manglietia species into 4 clades based on chloroplast genomes and clarify the interspecific relationships between M. longirostrata and M.hookeri, as well as between M. guangnanica and M. calcarea. This enriches the cp genome data of Manglietia, provides new insights into infrageneric classification, and lays a foundation for further phylogenetic and evolutionary studies of Manglietia.

comments1: Title: Overall, the title of the manuscript is clear. However, since the study involves three species, it would be more appropriate to use the plural form. Regarding nomenclature, please ensure that the genus Manglietia is italicized throughout the document, in accordance with standard taxonomic conventions. Suggested revised title: “The Complete Chloroplast Genomes of Three Manglietia Species and Phylogenetic Insights into the Genus Manglietia Blume”

response1: Thank you so much. Based on your feedback, we have revised the title from “Genome” to ‘Genomes’ and italicized “Manglietia” throughout the text.

comments2: Line 43: It is recommended to define the abbreviation at its first mention, for example: “the chloroplast (cp) genome.”

response2: Thank you very much for your suggestions. We have provided the full name upon the first occurrence of the abbreviation, as you suggested.

comments3: Line 48: Please use the abbreviation “cp genome” after the first mention of “chloroplast genome.”

response3: Thanks. As your suggestion, We have uniformly modified the recurring term “chloroplast genome” to “cp genome”.

comments4: Line 53: The authors should use abbreviations after the first full mention. Please review and apply this throughout the manuscript (e.g., to increasingly favor the use of cp genomes).

response4: Thanks to your advice, we have conducted a full-text verification and made revisions.

comments5: Line 72: The full name of the genus Manglietia is appropriately introduced at the beginning. To avoid redundancies and improve readability, you may use the abbreviated form of the genus, as long as it does not cause ambiguity. (e.g., M. fordiana).

response5: Thanks a lot. As your suggestion, we have abbreviated the genus names of subsequent species.

comments6: Line 131-132: The authors should use a consistent font style and size throughout all sections, including references to databases and URLs.

response6: Thanks. Based on your feedback, we have reviewed and adjusted the font size and formatting throughout the entire document.

comments7: Line 152: It is no longer necessary to mention the full genus name, even when referring to a different species. Please review and apply this throughout the manuscript (e.g., The cp genomes of M. guangnanica, M. longirostrata, and M. hookeri).

response7: Thanks a lot. As your suggestion, we have abbreviated the genus names for species belonging to the same genus.

comments8: Line 155: There appears to be an error in the reference management. Authors should review and update cross-references to ensure they are correctly linked, especially to figures and tables.

response8: Thank you so much. Based on your feedback, we have re-cross-referenced all Figure and tables throughout the document.

comments9: Line 168: The figure caption lacks sufficient detail to fully describe the results. While it may seem redundant, a complete and accurate figure description is necessary for clarity and proper interpretation.

response9: Thanks a lot. As your suggestion, we have added new captions to fully describe the results. Added content is displayed in red font.

comments10: Line 175: Table1. The column headers in the table are not centered. Please adjust the alignment to ensure consistency and improve the table's readability.

response10: Thanks to your advice, we have revised the column headers.

comments11: Line 191-192: There appears to be the same reference error mentioned above, likely caused by the reference manager. It seems the authors intended to refer to Figure 2, but the current placeholder "(Error! Reference source not found.)" Is confusing and disrupts the flow of the text.

response11: Thank you so much. Based on your feedback, we have re-cross-referenced all Figure and tables throughout the document.

comments12: Line 194: The figure caption is too general. While it may seem redundant, a complete and accurate figure description is necessary for clarity and proper interpretation.

response12: Thanks a lot. As your suggestion, we have added new captions to fully describe the results. Added content is displayed in red font.

comments13: Line 356: Based on the objective analysis, what is the biological or evolutionary significance of the distribution of Manglieta across the different clade?

response13: Thanks a lot. As your suggestion, we have added some branch information to the figure captions.

comments14: Line 532: I recommend repositioning either the figure or its initial reference to improve clarity, as its current placement gives the impression that it is disconnected from the content. If the figure is not essential to the main text, consider moving it to the supplementary materials.

response14: Thanks. Based on your feedback, We have rearranged the position of the figures throughout the text to align them more closely with the content.

comments15: Line 542: The authors should indicate how the integrity of the DNA was assessed (e.g., agarose gel electrophoresis).

response15: Thanks. As your suggestion, we have indicate methods for detecting DNA integrity. Added content is displayed in red font.

comments16: Line 546: The subtitle on line 542 is repeated.

response16: Thank you so much. Based on your feedback, we have changed the subtitle here to “Chloroplast Genome Assembly and Annotation”.

comments17: Line 547: The authors should include additional information such as the type of library, read length, amount of data generated, and the parameters used during data filtering.

response17: Thanks a lot. As your suggestion, We supplemented relevant information such as type of library, read length, and amount of data generated. Additional information is displayed in red font.

comments18: Line 548: It is important to mention the parameters used for genome assembly or whether default settings were applied. Also indicate the total number of reads used or the size of the input file.

response18: Thanks a lot. As your suggestion, we supplemented the parameters used during genome assembly. Supplementary information is displayed in red font.

comments19: Line 554: What version of OGDRAW did you use to generate the chloroplast genome maps?

response19: Thank you so much. Based on your feedback, we have added OGDRAW version information.

comments20: Line 555: Preferably add the software versions please.

response20: Thanks a lot. Based on your feedback, we have added version information for the software used in the article. The online tool provides the URL and access date.

comments21: Line 567: The authors should use a consistent font style and size.

response21: Thanks. Based on your feedback, We have reviewed and adjusted the font size and formatting in the article.

comments22: For EMBOSS the authors do not specify the version, it would be useful to add it if known.

response22: Thank you so much. Based on your feedback, we have added EMBOSS version information.

comments23: Line 571-572: Briefly explain what these analyses measure.

response23: Thanks a lot. As your suggestion, we have added some content to clarify the measurement objectives of these analytical methods. Added content is displayed in red font.

comments24: Line 589: “KaKs_Calculator” is mentioned, but neither the version nor the source (link or source) are indicated. This is important for reproducibility.

response24: Thank you so much. Based on your feedback, we have added KaKs_Calculator version information.

comments25: Line 631: The authors make an important taxonomic claim: “this study considers M. guangnanica and M. calcarea to be two distinct species.” However, sufficient phylogenetic support values to substantiate this conclusion are not provided. Moreover, the level of genomic divergence between these taxa is not discussed.

response25: Thanks a lot. As your suggestion, we have added some content (For example, phylogenetic support values and CP genome similarity, etc.) to support the important taxonomic claim: “this study considers M. guangnanica and M. calcarea to be two distinct species.” Added content is displayed in red font.

comments26: Line 634: The supplementary tables mentioned were not found, except for Table S6. It is recommended to verify that all supplementary materials are properly included. The genus names of species should be written correctly and italicized.

response26: Thank you so much. Based on your feedback, we have modified the genus names of species. The supplementary tables comprise a total of six tables, designated as Table S1 ~ Table S6. The lower-left corner allows you to switch between viewing the contents of each table.

comments27: Line 649: This should also be included in the Materials and Methods section.

response27: Thanks a lot. As your suggestion, we added the NCBI accession numbers for the three species in the Materials and Methods section. Added content is displayed in red font.

comments28: Line 654: The reference list needs careful revision. First, the formatting is inconsistent, with a mixture of citation styles and font sizes. Please ensure that the reference formatting is standardized according to the journal’s guidelines and that DOIs are provided for all references where applicable. The DOI in references is recommended to be written in its complete form as a URL link: https://doi:10.1093/pcp/pcn069.

response28: Thanks. Based on your feedback, we have standardized the format of the references and supplemented the DOI for the literature according to the journal's guidelines.

comments29: Line 801-line 804: The paragraph is repeated again. It is recommended to carefully review the manuscript to eliminate unnecessary repetitions and improve the clarity of the text.

response29: Thank you so much. Based on your feedback, we have eliminate the paragraph.

Reviewer 3 Report

Comments and Suggestions for Authors

Dear colleagues, put the links in order starting from line 155.
Line 175. Table 1. Characteristics of the complete chloroplast genomes of the three Manglietia species.
Please put the table in order.

Author Response

Response to Referee 3

The article ‘The Complete Chloroplast Genome of Three Manglietia Species and Phylogenetic Insight into the Genus Manglietia Blume’ by Luo et al. The aim was to clarify Classification system and interspecific relationships within the Manglietia genus. In this study, the authors classified 28 Manglietia species into 4 clades based on chloroplast genomes and clarify the interspecific relationships between M. longirostrata and M.hookeri, as well as between M. guangnanica and M. calcarea. This enriches the cp genome data of Manglietia, provides new insights into infrageneric classification, and lays a foundation for further phylogenetic and evolutionary studies of Manglietia.

comments1: Dear colleagues, put the links in order starting from line 155.

response1: Thank you so much. Based on your feedback, we have updated cross-references for figures and tables throughout the article.

comments2: Line 175. Table 1. Characteristics of the complete chloroplast genomes of the three Manglietia species.Please put the table in order.

response2: Thanks a lot. As your suggestion, we have rearranged Table 1 to make the results clearer.

comments3: Are the results clearly presented?  It must be improved.

response3: Thanks. Based on your feedback, we have rearranged the tables and figures in this article and enhanced the image resolution to improve the clarity of the research findings.

Reviewer 4 Report

Comments and Suggestions for Authors

The authors of the manuscript “The Complete Chloroplast Genome of Three Manglietia Species and Phylogenetic Insight into the Genus Manglietia Blume” have sequenced, assembled, and annotated the chloroplast genomes of three Manglietia species and provided detailed structural analyses of these genomes. Their phylogenetic analyses divide Manglietia into four clades, differing from previously proposed infrageneric classification systems. The authors recognize M. guangnanica and M. calcarea as two distinct species, and their findings do not support treating M. longirostrata as a variety of M. hookeri.

Some remarks:

Figure 2: The caption states, “Different colors represent repeats of different lengths.” The authors should specify the unit of measurement for length.

Figure 5: There do not appear to be discernible differences between parts a, b, and c of the figure. If differences exist, they are not visible due to poor figure quality. If there are no differences, please consider omitting the redundant panels.

The accession numbers of the sequenced chloroplast genomes are only mentioned in Table 1. It is advisable to include an explicit statement within the main text describing the annotated sequences with their accession numbers and to indicate the database in which they are deposited.

Under accession number PQ246290 in the NCBI database, the annotation is Magnolia fordiana var. calcarea, whereas in the article, it is referred to as Manglietia guangnanica. Please clarify this discrepancy within the manuscript. Similarly, accession number PV239601 in the NSBI database is annotated as Magnolia hookeri var. longirostrata, whereas in the article, it is cited as Manglietia longirostrata. Please make it clear to the readers why this discrepancy occurs.

Author Response

Response to Referee 4

The article ‘The Complete Chloroplast Genome of Three Manglietia Species and Phylogenetic Insight into the Genus Manglietia Blume’ by Luo et al. The aim was to clarify Classification system and interspecific relationships within the Manglietia genus. In this study, the authors classified 28 Manglietia species into 4 clades based on chloroplast genomes and clarify the interspecific relationships between M. longirostrata and M.hookeri, as well as between M. guangnanica and M. calcarea. This enriches the cp genome data of Manglietia, provides new insights into infrageneric classification, and lays a foundation for further phylogenetic and evolutionary studies of Manglietia.

comments1: Figure 2: The caption states, “Different colors represent repeats of different lengths.” The authors should specify the unit of measurement for length.

response1: Thank you very much for your suggestions. We have specified the units of measurement used for length. Added content is displayed in red font.

comments2: Figure 5: There do not appear to be discernible differences between parts a, b, and c of the figure. If differences exist, they are not visible due to poor figure quality. If there are no differences, please consider omitting the redundant panels.

response2: Thank you so much. Based on your feedback, we have enhanced the quality of Figure 5 and labeled the RSCU values for each codon in the figure to more clearly illustrate the differences in results.

comments3: The accession numbers of the sequenced chloroplast genomes are only mentioned in Table 1. It is advisable to include an explicit statement within the main text describing the annotated sequences with their accession numbers and to indicate the database in which they are deposited.

response3: Thanks a lot. As your suggestion, we have added accession numbers and other information for the chloroplast genome sequences of three species in the Materials and Methods section. Added content is displayed in red font.

comments4: Under accession number PQ246290 in the NCBI database, the annotation is Magnolia fordianavar. calcarea, whereas in the article, it is referred to as Manglietia guangnanica. Please clarify this discrepancy within the manuscript. Similarly, accession number PV239601 in the NSBI database is annotated as Magnolia hookeri var. longirostrata, whereas in the article, it is cited as Manglietia longirostrata. Please make it clear to the readers why this discrepancy occurs.

response4: Thank you very much for your suggestions. As mentioned in the article, the interspecific relationships among some species within the genus Manglietia remain ambiguous, leading to unclear nomenclature for these species in the NCBI database. When submitting data, we inquired about this issue with NCBI officials, but they designated M. guangnanica as Magnolia fordiana var. calcarea based on the NCBI Taxonomy. Thank you once again for your thorough review. We have briefly outlined these circumstances in the Introduction. Added content is displayed in red font.

Reviewer 5 Report

Comments and Suggestions for Authors

The paper has a great scientific significance. On the other hand, there are some issues that need to be addressed. First of all, I am afraid that abstract is too long and could be briefly written. Similar can be applied to the introductory part. This chapter should focus just to the most important things related to the topic. You put Results before Materials and Methods. Is it in accordance to the journal rules, because more logical is to describe materials and methods and then consider the obtained results? As for the results, there are many places where you put Error! Reference source not found. I am not sure that I understood what you wanted to say. If it is your result, you don't need to use the reference. That would be meaningful in the Discussion chapter. Would you be so kind as to explain me what it was about? In the Materials and Methods chapter, you mentioned subsection Sample selection, DNA extraction and sequencing two times. Did you use two different methods for the same thing? If not, why did you separate it? Conclusions might be a bit more connected with the obtained results. All in all, the paper is very good and you should make some small revisions as I wrote. 

Author Response

Response to Referee 5

The article ‘The Complete Chloroplast Genome of Three Manglietia Species and Phylogenetic Insight into the Genus Manglietia Blume’ by Luo et al. The aim was to clarify Classification system and interspecific relationships within the Manglietia genus. In this study, the authors classified 28 Manglietia species into 4 clades based on chloroplast genomes and clarify the interspecific relationships between M. longirostrata and M.hookeri, as well as between M. guangnanica and M. calcarea. This enriches the cp genome data of Manglietia, provides new insights into infrageneric classification, and lays a foundation for further phylogenetic and evolutionary studies of Manglietia.

comments1: First of all, I am afraid that abstract is too long and could be briefly written. Similar can be applied to the introductory part. This chapter should focus just to the most important things related to the topic.

response1: Thank you very much for your suggestions. We have abbreviated some repetitive terminology in the abstract and Introduction sections and eliminated unnecessary content to prevent excessive redundancy in these parts.

comments2: You put Results before Materials and Methods. Is it in accordance to the journal rules, because more logical is to describe materials and methods and then consider the obtained results?

response2: Thank you so much. Based on your feedback, we have placed the “Materials and Methods” section before the “Results” section in accordance with journal rules to enhance the logical flow of this article.

comments3: As for the results, there are many places where you put Error! Reference source not found. I am not sure that I understood what you wanted to say. If it is your result, you don't need to use the reference. That would be meaningful in the Discussion chapter. Would you be so kind as to explain me what it was about?

response3: Thank you for your feedback. The errors in the figures and tables within this article may have been caused by issues with cross-referencing during document transmission. We have re-established cross-references for the tables and images in the article to ensure clearer presentation of the results. Based on your suggestions, we found that the references in the results section were excessive. We have removed most of the references in the results section, retaining only those that help us better present the findings.

comments4: In the Materials and Methods chapter, you mentioned subsection Sample selection, DNA extraction and sequencing two times. Did you use two different methods for the same thing? If not, why did you separate it?

response4: Thank you so much. Based on your feedback, we have changed the subtitle here to “Chloroplast Genome Assembly and Annotation”.

comments5: Conclusions might be a bit more connected with the obtained results.

response5: Thank you very much for your suggestions. We have added some content to establish a closer connection between the conclusion section and the results obtained. Added content is displayed in red font.
